# ENTROPIC GRADIENT DESCENT ALGORITHMS AND WIDE FLAT MINIMA

**Fabrizio Pittorino**[1,2]**, Carlo Lucibello**[1]**, Christoph Feinauer**[1]**,
Gabriele Perugini**[1]**, Carlo Baldassi**[1]**, Elizaveta Demyanenko**[1]**,
Riccardo Zecchina**[1]

[1]AI Lab, Institute for Data Science and Analytics, Bocconi University,
20136 Milano, Italy
[2]Dept. Applied Science and Technology, Politecnico di Torino, 10129 Torino, Italy

## ABSTRACT

The properties of flat minima in the empirical risk landscape of neural networks have been debated for some time. Increasing evidence suggests they possess better generalization capabilities with respect to sharp ones. In this work we first discuss the relationship between alternative measures of flatness: The *local entropy*, which is useful for analysis and algorithm development, and the *local energy*, which is easier to compute and was shown empirically in extensive tests on state-of-the-art networks to be the best predictor of generalization capabilities. We show semi-analytically in simple controlled scenarios that these two measures correlate strongly with each other and with generalization. Then, we extend the analysis to the deep learning scenario by extensive numerical validations. We study two algorithms, Entropy-SGD and Replicated-SGD, that explicitly include the local entropy in the optimization objective. We devise a training schedule by which we consistently find flatter minima (using both flatness measures), and improve the generalization error for common architectures (e.g. ResNet, EfficientNet).

## 1 INTRODUCTION

The geometrical structure of the loss landscape of neural networks has been a key topic of study for several decades (Hochreiter & Schmidhuber, 1997; Keskar et al., 2016). One area of ongoing research is the connection between the flatness of minima found by optimization algorithms like stochastic gradient descent (SGD) and the generalization performance of the network (Baldassi et al., 2020; Keskar et al., 2016). There are open conceptual problems in this context: On the one hand, there is accumulating evidence that flatness is a good predictor of generalization (Jiang et al., 2019). On the other hand, modern deep networks using ReLU activations are invariant in their outputs with respect to rescaling of weights in different layers (Dinh et al., 2017), which makes the mathematical picture complicated[1]. General results are lacking. Some initial progress has been made in connecting PAC-Bayes bounds for the generalization gap with flatness (Dziugaite & Roy, 2018).

The purpose of this work is to shed light on the connection between flatness and generalization by using methods and algorithms from the statistical physics of disordered systems, and to corroborate the results with a performance study on state-of-the-art deep architectures.

Methods from statistical physics have led to several results in the last years. Firstly, wide flat minima have been shown to be a structural property of shallow networks. They exist even when training on random data and are accessible by relatively simple algorithms, even though coexisting with exponentially more numerous minima (Baldassi et al., 2015; 2016a;

---

[1]We note, in passing, that an appropriate framework for theoretical studies would be to consider networks with binary weights, for which most ambiguities are absent.

2020). We believe this to be an overlooked property of neural networks, which makes them particularly suited for learning. In analytically tractable settings, it has been shown that flatness depends on the choice of the loss and activation functions, and that it correlates with generalization (Baldassi et al., 2020; 2019).

In the above-mentioned works, the notion of flatness used was the so-called *local entropy* (Baldassi et al., 2015; 2016a). It measures the low-loss volume in the weight space around a minimizer, as a function of the distance (i.e. roughly speaking it measures the amount of "good" configurations around a given one). This framework is not only useful for analytical calcuations, but it has also been used to introduce a variety of efficient learning algorithms that focus their search on flat regions (Baldassi et al., 2016a; Chaudhari et al., 2019; 2017). In this paper we call them *entropic algorithms*.

A different notion of flatness, that we refer to as *local energy* in this paper, measures the average profile of the training loss function around a minimizer, as a function of the distance (i.e. it measures the typical increase in the training error when moving away from the minimizer). This quantity is intuitively appealing and rather easy to estimate via sampling, even in large systems. In Jiang et al. (2019), several candidates for predicting generalization performance were tested using an extensive numerical approach on an array of different networks and tasks, and the local energy was found to be among the best and most consistent predictors.

The two notions, local entropy and local energy, are distinct: in a given region of a complex landscape, the local entropy measures the size of the lowest valleys, whereas the local energy measures the average height. Therefore, in principle, the two quantities could vary independently. It seems reasonable, however, to conjecture that they would be highly correlated under mild assumptions on the roughness of the landscape (which is another way to say that they are both reasonable measures to express the intuitive notion of "flatness").

In this paper, we first show that for simple systems in controlled conditions, where all relevant quantities can be estimated well by using the Belief Propagation (BP) algorithm (Mezard & Montanari (2009)), the two notions of flatness are strongly correlated: regions of high local entropy have low local energy, and vice versa. We also confirm that they are both correlated with generalization.

This justifies the expectation that, even for more complex architectures and datasets, those algorithms which are driven towards high-local-entropy regions would minimize the local energy too, and thus (based on the findings in Jiang et al. (2019)) would find minimizers that generalize well. Indeed, we systematically applied two entropic algorithms, Entropy-SGD (eSGD) and Replicated-SGD (rSGD), to state-of-the-art deep architectures, and found that we could achieve an improved generalization performance, at the same computational cost, compared to the original papers where those architectures were introduced. We believe these results to be an important addition to the current state of knowledge, since in (Baldassi et al. (2016b)) rSGD was applied only to shallow networks with binary weights trained on random patterns and the current work represents the first study of rSGD in a realistic deep neural network setting. Together with the first reported consistent improvement of eSGD over SGD on image classification, these results point to a very promising direction for further research. While we hope to foster the application of entropic algorithms by publishing code that can be used to adapt them easily to new architectures, we also believe that the numeric results are important for theoretical research, since they are rooted in a well-defined geometric interpretation of the loss landscape.

We also confirmed numerically that the minimizers found in this way have a lower local energy profile, as expected. Remarkably, these results go beyond even those where the eSGD and rSGD algorithms were originally introduced, thanks to a general improvement in the choice for the learning protocol, that we also discuss; apart from that, we used little to no hyper-parameter tuning.

## 2 RELATED WORK

The idea of using the flatness of a minimum of the loss function, also called the *fatness of the posterior* and the *local area estimate of quality*, for evaluating different minimizers is several decades old (Hochreiter & Schmidhuber, 1997; Hinton & van Camp, 1993; Buntine & Weigend, 1991). These works connect the flatness of a minimum to information theoretical concepts like the *minimum description length* of its minimizer: flatter minima correspond to minimizers that can be encoded using fewer bits. For neural networks, a recent empirical study (Keskar et al., 2016) shows that large-batch methods find sharp minima while small-batch ones find flatter ones, with a positive effect on generalization performance.

PAC-Bayes bounds can be used for deriving generalization bounds for neural networks (Zhou et al., 2018). In Dziugaite & Roy (2017), a method for optimizing the PAC-Bayes bound directly is introduced and the authors note similarities between the resulting objective function and an objective function that searches for flat minima. This connection is further analyzed in Dziugaite & Roy (2018).

In Jiang et al. (2019), the authors present a large-scale empirical study of the correlation between different complexity measures of neural networks and their generalization performance. The authors conclude that PAC-Bayes bounds and flatness measures (in particular, what we call local energy in this paper) are the most predictive measures of generalization.

The concept of local entropy has been introduced in the context of a statistical mechanics approach to machine learning for discrete neural networks in Baldassi et al. (2015), and subsequently extended to models with continuous weights. We provide a detailed definition in the next section, but mention here that it measures a volume in the space of configurations, which poses computational difficulties. On relatively tractable shallow networks, the local entropy of any given configuration can be computed efficiently using Belief Propagation, and it can be also used directly as a training objective. In this setting, detailed analytical studies accompanied by numerical experiments have shown that the local entropy correlates with the generalization error and the eigenvalues of the Hessian (Baldassi et al., 2015; 2020). Another interesting finding is that the cross-entropy loss (Baldassi et al., 2020) and ReLU transfer functions (Baldassi et al., 2019), which have become the de-facto standard for neural networks, tend to bias the models towards high local entropy regions (computed based on the error loss).

Extending such techniques for general architectures is an open problem. However, the local entropy objective can be approximated to derive general algorithmic schemes. Replicated stochastic gradient descent (rSGD) replaces the local entropy objective by an objective involving several replicas of the model, each one moving in the potential induced by the loss while also attracting each other. The method has been introduced in Baldassi et al. (2016a), but only demonstrated on shallow networks. The rSGD algorithm is closely related to Elastic Averaging SGD (EASGD), presented in Zhang et al. (2014), even though the latter was motivated purely by the idea of enabling massively parallel training and had no theoretical basis. The substantial distinguishing feature of rSGD compared to EASGD when applied to deep networks is the focusing procedure, discussed in more detail below. Another difference is that in rSGD there is no explicit master replica.

Entropy-SGD (eSGD), introduced in Chaudhari et al. (2019), is a method that directly optimizes the local entropy using stochastic gradient Langevin dynamics (SGLD) (Welling & Teh, 2011). While the goal of this method is the same as rSGD, the optimization techniques involves a double loop instead of replicas. Parle (Chaudhari et al., 2017), combines eSGD and EASGD (with added focusing) to obtain a distributed algorithm that shows also excellent generalization performance, consistently with the results obtained in this work.

## 3 FLATNESS MEASURES: LOCAL ENTROPY, LOCAL ENERGY

The general definition of the local entropy loss $\mathcal{L}_{\mathrm{LE}}$ for a system in a given configuration $w$ (a vector of size $N$) can be given in terms of any common (usually, data-dependent) loss $\mathcal{L}$

as:

$$\mathcal{L}_{\text{LE}}(w) = -\frac{1}{\beta} \log \int dw' \; e^{-\beta \mathcal{L}(w') - \beta \gamma \text{d}(w', w)}. \tag{1}$$

The function d measures a distance and is commonly taken to be the squared norm of the difference of the configurations $w$ and $w'$:

$$\text{d}(w', w) = \frac{1}{2} \sum_{i=1}^{N} (w'_i - w_i)^2 \tag{2}$$

The integral is performed over all possible configurations $w'$; for discrete systems, it can be substituted by a sum. The two parameters $\beta$ and $\tilde{\gamma} = \beta\gamma$ are Legendre conjugates of the loss and the distance. For large systems, $N \gg 1$, the integral is dominated by configurations having a certain loss value $\mathcal{L}^*(w, \beta, \gamma)$ and a certain distance $\text{d}^*(w, \beta, \gamma)$ from the reference configuration $w$. These functional dependencies can be obtained by a saddle point approximation. In general, increasing $\beta$ reduces $\mathcal{L}^*$ and increasing $\tilde{\gamma}$ reduces $\text{d}^*$.

While it is convenient to use Eq. (1) as an objective function in algorithms and for the theoretical analysis of shallow networks, it is more natural to use a normalized definition with explicit parameters when we want to measure the flatness of a minimum. We thus also introduce the normalized local entropy $\Phi_{\text{LE}}(w, d)$, which, for a given configuration $w \in \mathbb{R}^N$, measures the logarithm of the volume fraction of configurations whose training error is smaller or equal than that of the reference $w$ in a ball of squared-radius $2d$ centered in $w$:

$$\Phi_{\text{LE}}(w, d) = \frac{1}{N} \log \frac{\int dw' \; \Theta(E_{\text{train}}(w) - E_{\text{train}}(w')) \, \Theta(d - \text{d}(w', w))}{\int dw' \; \Theta(d - \text{d}(w', w))}. \tag{3}$$

Here, $E_{\text{train}}(w)$ is the error on the training set for a given configuration $w$ and $\Theta(x)$ is the Heaviside step function, $\Theta(x) = 1$ if $x \geq 0$ and 0 otherwise. This quantity is upper-bounded by zero and tends to zero for $d \to 0$ (since for almost any $w$, except for a set with null measure, there is always a sufficiently small neighborhood in which $E_{\text{train}}$ is constant). For sharp minima, it is expected to drop rapidly with $d$, whereas for flat regions it is expected to stay close to zero within some range.

A different notion of flatness is that used in Jiang et al. (2019), which we call local energy. Given a weight configuration $w \in \mathbb{R}^N$, we define $\delta E_{\text{train}}(w, \sigma)$ as the average training error difference with respect to $E_{\text{train}}(w)$ when perturbing $w$ by a (multiplicative) noise proportional to a parameter $\sigma$:

$$\delta E_{\text{train}}(w, \sigma) = \mathbb{E}_z \, E_{\text{train}}(w + \sigma z \odot w) - E_{\text{train}}(w), \tag{4}$$

where $\odot$ denotes the Hadamard (element-wise) product and the expectation is over normally distributed $z \sim \mathcal{N}(0, I_N)$. In Jiang et al. (2019), a single, arbitrarily chosen value of $\sigma$ was used, whereas we compute entire profiles within some range $[0, \sigma_{\max}]$ in all our tests.

## 4 ENTROPIC ALGORITHMS

For our numerical experiments we have used two entropic algorithms, rSGD and eSGD, mentioned in the introduction. They both approximately optimize the local entropy loss $\mathcal{L}_{\text{LE}}$ as defined in Eq. (1), for which an exact evaluation of the integral is in the general case intractable. The two algorithms employ different but related approximation strategies.

**Entropy-SGD.** Entropy-SGD (eSGD), introduced in Chaudhari et al. (2019), minimizes the local entropy loss Eq. (1) by approximate evaluations of its gradient. The gradient can be expressed as

$$\nabla \mathcal{L}_{\text{LE}}(w) = \gamma(w - \langle w' \rangle) \tag{5}$$

where $\langle \cdot \rangle$ denotes the expectation over the measure $Z^{-1} e^{-\beta \mathcal{L}(w') - \beta \gamma \text{d}(w', w)}$, where $Z$ is a normalization factor. The eSGD strategy is to approximate $\langle w' \rangle$ (which implicitly depends on $w$) using $L$ steps of stochastic gradient Langevin dynamics (SGLD). The resulting double-loop algorithm is presented as Algorithm 1. The noise parameter $\epsilon$ in the algorithm is linked to

**Algorithm 1:** Entropy-SGD (eSGD)

| | |
|---|---|
| **Input** | : $w$ |
| **Hyper-parameters** | : $L, \eta, \gamma, \eta', \epsilon, \alpha$ |

1  **for** $t = 1, 2, \ldots$ **do**
2  $\quad$ $w', \mu \leftarrow w$
3  $\quad$ **for** $l = 1, \ldots, L$ **do**
4  $\quad\quad$ $\Xi \leftarrow$ sample minibatch
5  $\quad\quad$ $dw' \leftarrow \nabla\mathcal{L}(w'; \Xi) + \gamma(w' - w)$
6  $\quad\quad$ $w' \leftarrow w' - \eta' dw' + \sqrt{\eta'}\,\epsilon\,\mathcal{N}(0, I)$
7  $\quad\quad$ $\mu \leftarrow \alpha\mu + (1 - \alpha)w'$
8  $\quad$ $w \leftarrow w - \eta(w - \mu)$

**Algorithm 2:** Replicated-SGD (rSGD)

| | |
|---|---|
| **Input** | : $\{w^a\}$ |
| **Hyper-parameters** | : $y, \eta, \gamma, K$ |

1  **for** $t = 1, 2, \ldots$ **do**
2  $\quad$ $\bar{w} \leftarrow \frac{1}{y}\sum_{a=1}^{y} w^a$
3  $\quad$ **for** $a = 1, \ldots, y$ **do**
4  $\quad\quad$ $\Xi \leftarrow$ sample minibatch
5  $\quad\quad$ $dw^a \leftarrow \nabla\mathcal{L}(w^a; \Xi)$
6  $\quad\quad$ **if** $t = 0 \mod K$ **then**
7  $\quad\quad\quad$ $dw^a \leftarrow dw^a + K\gamma(w^a - \bar{w})$
8  $\quad\quad$ $w^a \leftarrow w^a - \eta\,dw^a$

the inverse temperature by the usual Langevin relation $\epsilon = \sqrt{2/\beta}$. In practice we always set it to the small value $\epsilon = 10^{-4}$ as in Chaudhari et al. (2019). For $\epsilon = 0$, eSGD approximately computes a proximal operator (Chaudhari et al., 2018). For $\epsilon = \alpha = \gamma = 0$, eSGD reduces to the recently introduced Lookahead optimizer (Zhang et al., 2019).

**Replicated-SGD.** Replicated-SGD (rSGD) consists in a replicated version of the usual stochastic gradient (SGD) method. In rSGD, a number $y$ of replicas of the same system, each with its own parameters $w_a$ where $a = 1, ..., y$, are trained in parallel for $K$ iterations. During training, they interact with each other indirectly through an attractive term towards their center of mass. As detailed in Baldassi et al. (2016a; 2020) in the simple case of shallow networks (committee machines), the replicated system, when trained with a stochastic algorithm such as SGD, collectively explores an approximation of the local entropy landscape without the need to explicitly estimate the integral in Eq. (1). In principle, the larger $y$ the better the approximation, but already with $y = 3$ the effect of the replication is significant. To summarize, rSGD replaces the local entropy Eq. (1) with the replicated loss $\mathcal{L}_R$:

$$\mathcal{L}_R(\{w^a\}_a) = \sum_{a=1}^{y} \mathcal{L}(w^a) + \gamma\sum_{a=1}^{y} \mathrm{d}(w^a, \bar{w}) \tag{6}$$

Here, $\bar{w}$ is a center replica defined as $\bar{w} = \frac{1}{y}\sum_{a=1}^{y} w^a$. The algorithm is presented as Algorithm 2. Thanks to focusing (see below), any of the replicas or the center $\bar{w}$ can be used after training for prediction. This procedure is parallelizable over the replicas, so that wall-clock time for training is comparable to SGD, excluding the communication which happens every $K$ parallel optimization steps. In order to decouple the communication period and the coupling hyperparameter $\gamma$, we let the coupling strength take the value $K\gamma$. In our experiments, we did not observe degradation in generalization performance with $K$ up to 10.

**Focusing.** A common feature of both algorithms is that the parameter $\gamma$ in the objective $\mathcal{L}_{\mathrm{LE}}$ changes during the optimization process. We start with a small $\gamma$ (targeting large regions and allowing a wider exploration of the landscape) and gradually increase it. We call this process *focusing*. Focusing improves the dynamics by driving the system quickly to wide regions and then, once there, gradually trading off the width in order to get to the minima of the loss within those regions, see Baldassi et al. (2016b;a). We adopt an exponential schedule for $\gamma$, where its value at epoch $\tau$ is given by $\gamma_\tau = \gamma_0(1 + \gamma_1)^\tau$. For rSGD, we fix $\gamma_0$ by balancing the distance and the data term in the objective before training starts, i.e. we set $\gamma_0 = \sum_a \mathcal{L}(w^a)/\sum_a \mathrm{d}(w^a, \bar{w})$ for rSGD. The parameter $\gamma_1$ is chosen such that $\gamma$ increases by a factor $10^4$. For eSGD, we were unable to find a criterion that worked for all experiments and manually tuned it.

**Optimizers.** Vanilla SGD updates in Algorithms 1 and 2 can be replaced by optimization steps of any commonly used gradient-based optimizers.

## 5   DETAILED COMPARISON OF FLATNESS MEASURES IN SHALLOW NETWORKS

In this section, we explore in detail the connection between the two flatness measures and the generalization properties in a one-hidden-layer network that performs a binary classification task, also called a committee machine. This model has a symmetry that allows to fix all the weights in the last layer to 1, and thus only the first layer is trained. It is also invariant to rescaling of the weights. This allows to study its typical properties analytically with statistical mechanics techniques, and it was shown in Baldassi et al. (2020) that it has a rich non-convex error-loss landscape, in which rare flat minima coexist with narrower ones. It is amenable to be studied semi-analytically: for individual instances, the minimizers found by different algorithms can be compared by computing their local entropy efficiently with the Belief Propagation (BP) algorithm (see Appendix B.1), bypassing the need to perform the integral in Eq. (1) explicitly. Doing the same for general architectures is an open problem.

For a network with $K$ hidden units, the output predicted for a given input pattern $x$ reads:

$$\hat{\sigma}(w, x) = \text{sign} \left[ \frac{1}{\sqrt{K}} \sum_{k=1}^{K} \text{sign} \left( \frac{1}{\sqrt{N}} \sum_{i=1}^{N} w_{ki} x_i \right) \right] \tag{7}$$

We follow the numerical setting of Baldassi et al. (2020) and train this network to perform binary classification on two classes of the Fashion-MNIST dataset with binarized patterns, comparing the results of standard SGD with cross-entropy loss (CE) with the entropic counterparts rSGD and eSGD. All these algorithms require a differentiable objective, thus we approximate sign activation functions on the hidden layer with $\tanh(\beta x)$ functions, where the $\beta$ parameter increases during the training. The CE loss is not invariant with respect to weight rescaling: we control the norm of the weights explicitly by keeping them normalized and introducing an overall scale parameter $\omega$ that we insert explicitly in the loss:

$$\mathcal{L}(w) = \mathbb{E}_{x,\sigma \sim D} \ f(\sigma \cdot \hat{\sigma}(w, x), \omega) \tag{8}$$

Here, we have defined $f(x, \omega) = -\frac{x}{2} + \frac{1}{2\omega} \log(2 \cosh(\omega x))$ as in Baldassi et al. (2020). The $\omega$ parameter is increased gradually in the training process in order to control the growth rate of the weight norms. Notice that the parameter $\beta$ could also be interpreted as a norm that grows over time.

As shown in Baldassi et al. (2020), slowing down the norm growth rate results in better generalization performance and increased flatness of the minima found at the end of the training. To appreciate this effect we used two different parameters settings for optimizing the loss in Eq.(8) with SGD, that we name "SGD slow" and "SGD fast". In the fast setting both $\beta$ and $\omega$ start with a large value and grow quickly, while in the slow setting they start from small values and grow more slowly, requiring more epochs to converge. For rSGD, we also used two different "fast" and "slow" settings, where the difference is in a faster or slower increase of the $\gamma$ parameter that controls the distance between replicas.

The results are shown in Fig. 1. In the left panel, we report $\Phi_{\text{LE}}$ computed with BP around the solutions found by the different algorithms, as a function of the distance from the solution. Even if the slow SGD setting improves the flatness of the solution found, entropy-driven algorithms are biased towards flatter minima, in the sense of the local entropy, as expected. In the central panel we plot the local energy profiles $\delta E_{\text{train}}$ for the same solutions, and we can see that the ranking of the algorithm is preserved: the two flatness measures agree. The same ranking is also clearly visible when comparing the generalization errors, in the right panel of the figure: flatter minima generalize better [2].

---

[2]In the appendix B.3 we show that the correlation between local entropy, local energy and generalization holds also in a setting where we do not explicitly increase the local entropy.

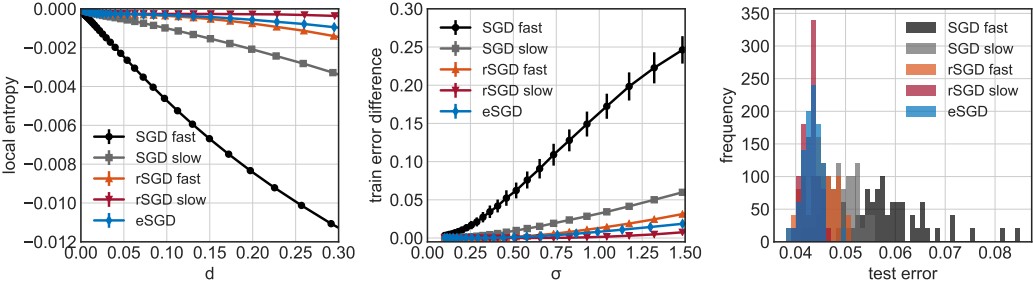

Figure 1: Normalized local entropy $\Phi_{\mathrm{LE}}$ as a function of the squared distance $d$ (left), training error difference $\delta E_{\mathrm{train}}$ as a function of perturbation intensity $\sigma$ (center) and test error distribution (right) for a committee machine as defined in Eq. 7, trained with various algorithms on the reduced version of the Fashion-MNIST dataset. Results are obtained using 50 random restarts for each algorithm.

# 6 NUMERICAL EXPERIMENTS ON DEEP NETWORKS

## 6.1 COMPARISONS ACROSS SEVERAL ARCHITECTURES AND DATASETS

In this section we show that, by optimizing the local entropy with eSGD and rSGD, we are able to systematically improve the generalization performance compared to standard SGD. We perform experiments on image classification tasks, using common benchmark datasets, state-of-the-art deep architectures and the usual cross-entropy loss. The detailed settings of the experiments are reported in the SM. For the experiments with eSGD and rSGD, we use the same settings and hyper-parameters (architecture, dropout, learning rate schedule,...) as for the baseline, unless otherwise stated in the SM and apart from the hyper-parameters specific to these algorithms. While it would be interesting to add weight normalization (Salimans & Kingma (2016)) with frozen norm, as we did for committee machine, none of the baselines that we compare against uses this method. We also note that for the local energy as defined in Eq. 4, the noise is multiplicative and local energy is norm-invariant if the model itself is norm-invariant.

While we do some little hyper-parameter exploration to obtain a reasonable baseline, we do not aim to reproduce the best achievable results with these networks, since we are only interested in comparing different algorithms in similar contexts. For instance, we train PyramidNet+ShakeDrop for 300 epochs, instead of the 1800 epochs used in Cubuk et al. (2018), and we start from random initial conditions for EfficientNet instead of doing transfer learning as done in Tan & Le (2019). In the case of the ResNet110 architecture instead, we use the training specification of the original paper (He et al., 2016).

| Dataset | Model | Baseline | rSGD | eSGD | rSGD×$y$ |
|---|---|---|---|---|---|
| **CIFAR-10** | SmallConvNet | $16.5 \pm 0.2$ | $15.6 \pm 0.3$ | $14.7 \pm 0.3$ | $14.9 \pm 0.2$ |
| | ResNet-18 | $13.1 \pm 0.3$ | $12.4 \pm 0.3$ | $12.1 \pm 0.3$ | $11.8 \pm 0.1$ |
| | ResNet-110 | $6.4 \pm 0.1$ | $6.2 \pm 0.2$ | $6.2 \pm 0.1$ | $5.3 \pm 0.1$ |
| | PyramidNet+ShakeDrop | $2.1 \pm 0.2$ | $2.2 \pm 0.1$ | | 1.8 |
| **CIFAR-100** | PyramidNet+ShakeDrop | $13.8 \pm 0.1$ | $13.5 \pm 0.1$ | | 12.7 |
| | EfficientNet-B0 | 20.5 | 20.6 | $20.1 \pm 0.2$ | 19.5 |
| **Tiny ImageNet** | ResNet-50 | $45.2 \pm 1.2$ | $41.5 \pm 0.3$ | $41.7 \pm 1$ | $39.2 \pm 0.3$ |
| | DenseNet-121 | $41.4 \pm 0.3$ | $39.8 \pm 0.2$ | $38.6 \pm 0.4$ | $38.9 \pm 0.3$ |

Table 1: Test set error (%) for vanilla SGD (baseline), eSGD and rSGD. The first three columns show results obtained with the same number of passes over the training data. In the last column instead, each replica in the parallelizable rSGD algorithm consumes the same amount of data as the baseline.

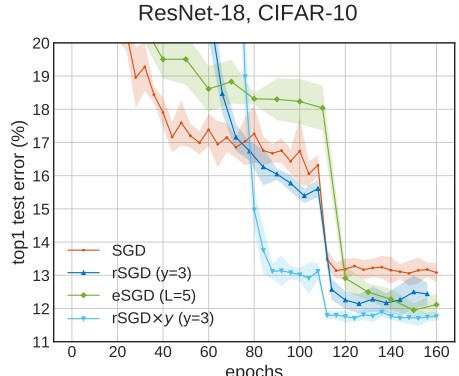
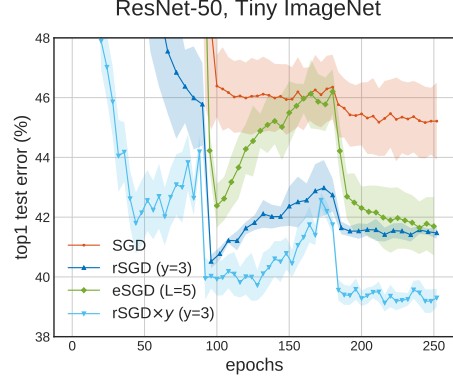

Figure 2: Left: Test error of ResNet-18 on CIFAR-10. Right: Test error of ResNet-50 on Tiny ImageNet. The curves are averaged over 5 runs. Training data consumed is the same for SGD, rSGD and eSGD. Epochs are rescaled by $y$ for rSGD and by $L$ for eSGD (they are not rescaled for rSGD$\times y$).

All combinations of datasets and architectures we tested are reported in Table 1, while representative test error curves are reported in Fig. 2. Blanks correspond to untested combinations. The first 3 columns correspond to experiments with the same number of effective epochs, that is considering that in each iteration of the outer loop in Algorithms 1 and 2 we sample $L$ and $y$ mini-batches respectively. In the last column instead, each replica consumes individually the same amount of data as the baseline. Being a distributable algorithm, rSGD enjoys the same scalability as the related EASGD and Parle (Zhang et al., 2014; Chaudhari et al., 2017).

For rSGD, we use $y = 3$ replicas and the scoping schedules described in Sec. 4. In our explorations, rSGD proved to be robust with respect to specific choices of the hyper-parameters. The error reported is that of the center replica $\bar{w}$. We note here, however, that the distance between the replicas at the end of training is very small and they effectively collapse to a single solution. Since training continues after the collapse and we reach a stationary value of the loss, we are confident that the minimum found by the replicated system corresponds to a minimum of a single system. For eSGD, we set $L = 5$, $\epsilon = 1e-4$ and $\alpha = 0.75$ in all experiments, and we perform little tuning for the the other hyper-parameters. The algorithm is more sensitive to hyper-parameters than rSGD, while still being quite robust. Moreover, it misses an automatic $\gamma$ scoping schedule.

Results in Table 1 show that entropic algorithms generally outperform the corresponding baseline with roughly the same amount of parameter tuning and computational resources. In the next section we also show that they end up in flatter minima.

## 6.2 Flatness vs generalization

For the deep network tests, we measured the local energy profiles (see Eq. (4)) of the configurations explored by the three algorithms. The estimates of the expectations were computed by averaging over 1000 perturbations for each value of $\sigma$. We did not limit ourselves to the end result, but rather we traced the evolution throughout the training and stopped when the training error and loss reached stationary values. In our experiments, the final training error is close to 0. Representative results are shown in Fig. 3, which shows that the eSGD and rSGD curves are below the SGD curve across a wide range of $\sigma$ values, while also achieving better generalization. Similar results are found for different architectures, as reported in Appendix B.3. This confirms the results of the shallow networks experiments: entropic algorithms tend to find flatter minima that generalize better, even when the hyper-parameters of the standard SGD algorithms had already been tuned for optimal generalization (and thus presumably to end up in generally flatter regions).

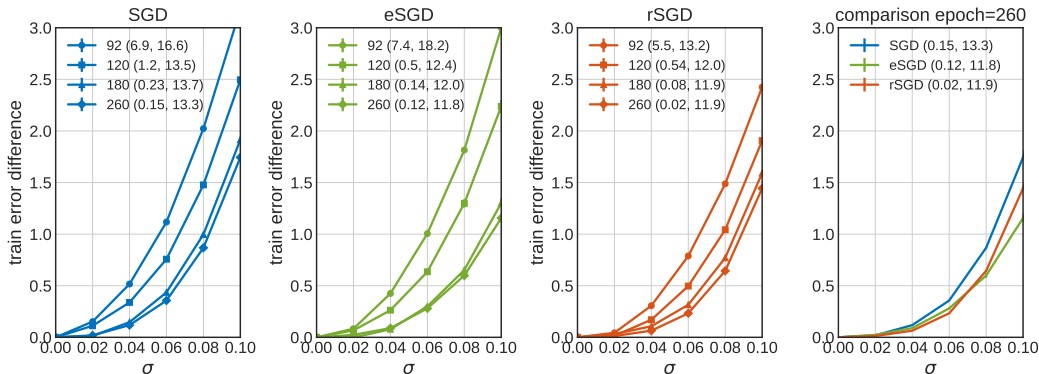

Figure 3: Evolution of the flatness along the training dynamics, for ResNet-18 trained on CIFAR-10 with different algorithms. Figures show the train error difference with respect to the unperturbed configurations. The value of the epoch, unperturbed train and test errors (%) are reported in the legends.The last panel shows that minima found at the end of an entropic training are flatter and generalize better. The value of the cross-entropy train loss of the final configurations is: 0.005 (SGD), 0.01 (eSGD), 0.005 (rSGD).

## 7 Discussion and conclusions

We studied the connection between two notions of flatness and generalization. We have performed detailed studies on shallow networks and an extensive numerical study on state of the art deep architectures. Our results suggest that local entropy is a good predictor of generalization performance. This is consistent with its relation to another flatness measure, the local energy, for which this property has already been established empirically. Furthermore, entropic algorithms can exploit this fact and be effective in improving the generalization performance on existing architectures, at fixed computational cost and with little hyper-parameter tuning. Our future efforts will be devoted to studying the connection between generalization bounds and the existence of wide flat regions in the landscape of the classifier.

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

## A  LOCAL ENTROPY AND REPLICATED SYSTEMS

The analytical framework of Local Entropy was introduced in Ref. Baldassi et al. (2015), while the connection between Local Entropy and systems of real replicas (as opposed to the "fake" replicas of spin glass theory (Mézard et al., 1987)) was made in Baldassi et al. (2016a). For convenience, we briefly recap here the simple derivation.

We start from the definition of the local entropy loss given in the main text:

$$\mathcal{L}_{\mathrm{LE}}\left(w\right) = -\frac{1}{\beta}\log\int dw'\ e^{-\beta\mathcal{L}\left(w'\right)-\frac{1}{2}\beta\gamma\|w'-w\|^2}. \tag{9}$$

We then consider the Boltzmann distribution of a system with energy function $\beta\mathcal{L}_{\mathrm{LE}}\left(w\right)$ and with an inverse temperature $y$, that is

$$p(w) \propto e^{-\beta y\mathcal{L}_{\mathrm{LE}}(w)}, \tag{10}$$

where equivalence is up to a normalization factor. If we restrict $y$ to integer values, we can then use the definition of $\mathcal{L}_{LE}$ to construct an equivalent but enlarged system, containing $y + 1$ replicas. Their joint distribution $p(w, \{w^a\}_a)$ is readily obtained by plugging Eq. (9) into Eq. (10). We can then integrate out the original configuration $w$ and obtain the marginal distributional for the $y$ remaining replicas

$$p(\{w^a\}_a) \propto e^{-\beta \mathcal{L}_R(\{w^a\}_a)}, \tag{11}$$

where the energy function is now given by

$$\mathcal{L}_R(\{w^a\}_a) = \sum_{a=1}^{y} \mathcal{L}(w^a) + \frac{1}{2}\gamma \sum_{a=1}^{y} \|w^a - \bar{w}\|^2, \tag{12}$$

with $\bar{w} = \frac{1}{y}\sum_a w^a$. We have thus recovered the loss function for the replicated SGD (rSGD) algorithm presented in the main text.

# B  FLATNESS AND LOCAL ENTROPY

## B.1  LOCAL ENTROPY ON THE COMMITTEE MACHINE

In what follows, we describe the details of the numerical experiments on the committee machine. We apply different algorithms to find zero error configurations and then use Belief Propagation (BP) to compute the local entropy curve for each configuration obtained. We compare them in Fig. 1, along with the local energy computed by sampling and their test errors.

We define a reduced version of the Fashion-MNIST dataset following Baldassi et al. (2020): we choose the classes Dress and Coat as they are non-trivial to discriminate but also different enough so that a small network as the one we used can generalize. The network is trained on a small subset of the available examples (500 patterns) binarized to $\pm 1$ by using the median of each image as a threshold on the inputs; we also filter both the training and test sets to use only images in which the median is between 0.25 and 0.75.

The network has input size $N = 784$ and a single hidden layer with $K = 9$ hidden units. The weights between the hidden layer and the output are fixed to 1. It is trained using mini-batches of 100 patterns. All the results are averaged over 50 independent restarts. For all algorithms we initialize the weights with a uniform distribution and then normalize the weights of the hidden units norm before the training starts and after each weight update. The $\beta$ and $\omega$ parameters are updated using exponential schedules, $\beta(t) = \beta_0 (1 + \beta_1)^t$ and $\omega(t) = \omega_0 (1 + \omega_1)^t$, where $t$ is the current epoch. An analogous exponential schedule is used for the elastic interaction $\gamma$ for rSGD and eSGD, as described in the main text. In the SGD fast case, we stop as soon as a solution with zero errors is found, while for SGD slow we stop when the cross entropy loss reaches a value lower than $10^{-7}$. For rSGD, we stop training as soon as the distance between the replicas and their center of mass is smaller than $10^{-8}$. For eSGD, we stop training as soon as the distance between the parameters and the mean ($\mu$ in Algorithm 1) is smaller than $10^{-8}$.

We used the following hyper-parameters for the various algorithms:

***SGD fast***: $\eta = 2 \cdot 10^{-4}$, $\beta_0 = 2.0$, $\beta_1 = 10^{-4}$, $\omega_0 = 5.0$, $\omega_1 = 0.0$;

***SGD slow***: $\eta = 3 \cdot 10^{-5}$, $\beta_0 = 0.5$, $\beta_1 = 10^{-3}$, $\omega_0 = 0.5$, $\omega_1 = 10^{-3}$;

***rSGD fast***: $\eta = 10^{-4}$, $y = 10$, $\gamma_0 = 2 \cdot 10^{-3}$, $\gamma_1 = 2 \cdot 10^{-3}$, $\beta_0 = 1.0$, $\beta_1 = 2 \cdot 10^{-4}$, $\omega_0 = 0.5$, $\omega_1 = 10^{-3}$;

***rSGD slow***: $\eta = 10^{-3}$, $y = 10$, $\gamma_0 = 10^{-4}$, $\gamma_1 = 10^{-4}$, $\beta_0 = 1.0$, $\beta_1 = 2 \cdot 10^{-4}$, $\omega_0 = 0.5$, $\omega_1 = 10^{-3}$;

***eSGD***: $\eta = 10^{-3}$, $\eta' = 5 \cdot 10^{-3}$, $\epsilon = 10^{-6}$, $L = 20$, $\gamma_0 = 10.0$, $\gamma_1 = 5 \cdot 10^{-5}$, $\beta_0 = 1.0$, $\beta_1 = 10^{-4}$, $\omega_0 = 0.5$, $\omega_1 = 5 \cdot 10^{-4}$;

For a given configuration $w$ obtained by one of the previous algorithms, we compute the local entropy through BP. The message passing involved is similar to the one accurately

detailed in Appendix IV of Ref. Baldassi et al. (2020), the only difference here being that we apply it to a fully-connected committee machine instead of the tree-like one used in Ref. Baldassi et al. (2020). We thus give here just a brief overview of the procedure. We frame the supervised learning problem as a constraint satisfaction problem given by the network architecture and the training examples. In order to compute the local entropy, we add a Gaussian prior centered in $w$ to the weights, $\mathcal{N}(w, \Delta)$. The variance $\Delta$ of the prior will be related later to a specific distance from $w$. We run BP iterations until convergence, then compute the Bethe free-energy $f_{Bethe}(w, \Delta)$ as a function of the fixed point messages. Applying the procedure for different values of $\Delta$ and then performing a Legendre transform, we finally obtain the local entropy curve $\Phi_{\mathrm{LE}}(w, d)$.

### B.2 Flatness curves for deep networks

In this Section we present flatness curves, $\delta E_{\mathrm{train}}(w, \sigma)$ from Eq. (4), for some of the deep networks architecture examined in this paper.

Results are reported in Figs 4 and 5 for different architectures and datasets. The expectation in Eq. (4) is computed over the complete training set using 100 and 400 realizations of the Gaussian noise for each data point in Figs 4 and 5 respectively. In experiments where data augmentation was used during training, it is also used when computing the flatness curve.

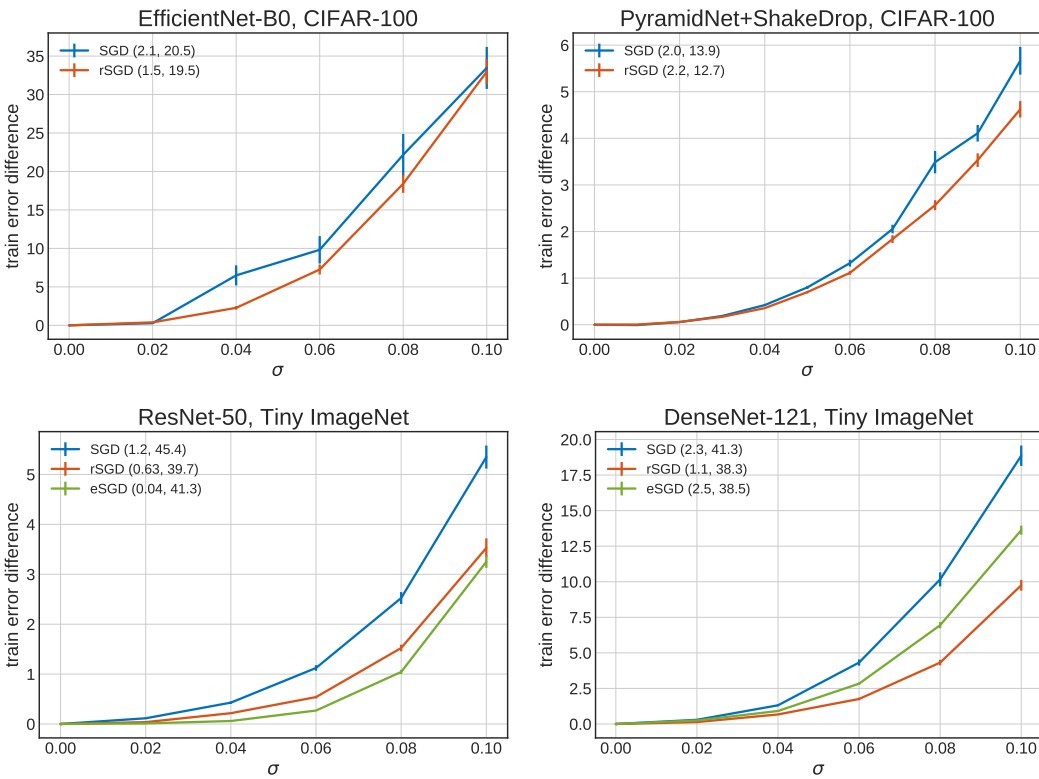

Figure 4: Train error difference $\delta E_{\mathrm{train}}$ from Eq. (4), for minina obtained on various architectures, datasets and with different algorithms, as a function of the perturbation intensity $\sigma$. Unperturbed train and test errors (%) are reported in the legends. The values of the train cross-entropy loss for the final configurations are: EfficientNet-B0: 0.08 (SGD), 0.06 (rSGD); PyramidNet 0.07 (SGD), 0.07 (rSGD): ResNet-50: 0.04 (SGD), 0.006 (eSGD), 0.1 (rSGD); DenseNet: 0.1 (SGD), 0.2 (eSGD), 0.12 (rSGD).

The comparison is performed between minima found by different algorithms, at a point where the training error is near zero and the loss has reached a stationary value. We note

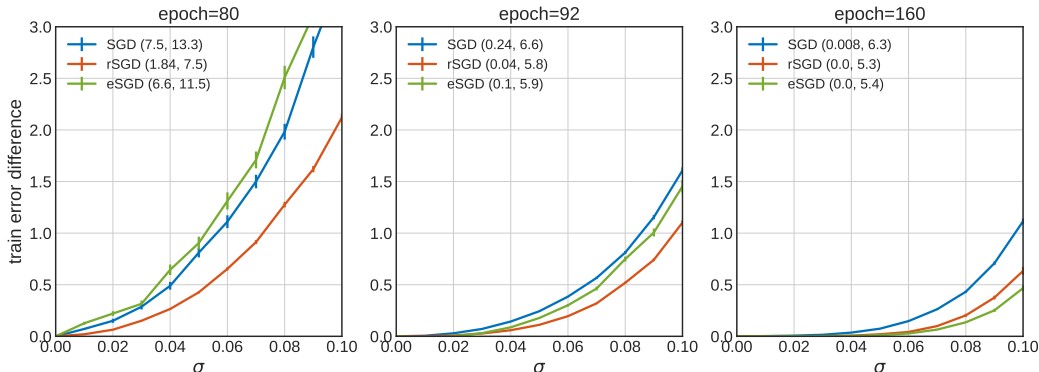

Figure 5: Train error difference $\delta E_{\text{train}}$ from Eq. (4) for ResNet-110 on Cifar-10. Values are computed along the training dynamics of different algorithms and as a function of the perturbation intensity $\sigma$. Unperturbed train and test errors (%) are reported in the legends. The values of the train cross-entropy loss for the final configurations are: 0.001 (SGD), 0.0005 (eSGD), 0.0005 (rSGD).

here that this kind of comparison is sensitive to the fact that the training errors at $\sigma = 0$ are close to each other. If the comparison is made for minima that show different train errors, the correlation between flatness and test error is not observed.

We report a generally good agreement between the flatness of the $\delta E_{\text{train}}$ curve and the generalization performance, for a large range of $\sigma$ values.

## B.3 CORRELATION OF FLATNESS AND GENERALIZATION

In this section we test more thoroughly the correlation of our definition of flatness with the generalization error. The entropic algorithms that we tested resulted in both increased flatness and increased generalization accuracy, but this does not imply that these quantities are correlated in other settings.

For the committee machine we tested in different settings that the local entropy provides the same information as the local energy, and that they both correlate well with the generalization error, even when entropic algorithms are not used. An example can be seen in Fig. 6, where the same architecture has been trained with SGD fast (see B.1) and different values of the dropout probability. The minima obtained with higher dropout have larger local entropy and generalize better.

In order to test this correlation independently in deep networks, we use models provided in the framework of the PGDL competition (see `https://sites.google.com/view/pgdl2020/`). These models have different generalization errors and have been obtained without using local entropy in their training objective.

We notice that the local entropy and the local energy are only comparable in the context of the same architecture, as the loss landscape may be very different if the network architecture varies. We therefore choose among the available models a subset with the same architecture (VGG-like with 3 convolutional blocks of width 512 and 1 fully connected layer of width 128) trained on CIFAR-10. Since the models were trained with different hyperparameters (dropout, batch size and weight decay), they still show a wide range of generalization errors.

Since we cannot compute the local entropy for these deep networks, we restrict the analysis to the computationally cheaper local energy (see Eq. 4) as a measure of flatness.

As shown in Fig. 7 we report a good correlation between flatness as measured by the train error difference (or local energy, at a given value of the perturbation intensity $\sigma$, namely

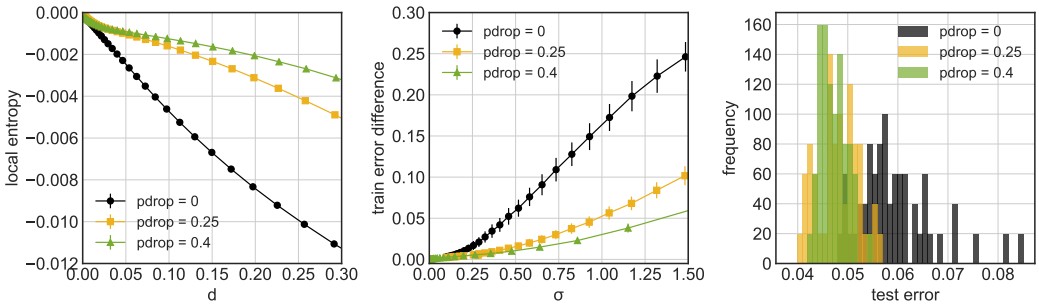

Figure 6: Normalized local entropy $\Phi_{\text{LE}}$ as a function of the squared distance $d$ (left), training error difference $\delta E_{\text{train}}$ as a function of perturbation intensity $\sigma$ (center) and test error distribution (right) for a committee machine trained with SGD fast and different dropout probabilities on the reduced version of the Fashion-MNIST dataset. Results are obtained using 50 random restarts for each algorithm.

$\sigma = 0.5$) and test error, resulting in a Pearson correlation coefficient $r(12) = 0.90$ with $p$-value 1e-4.

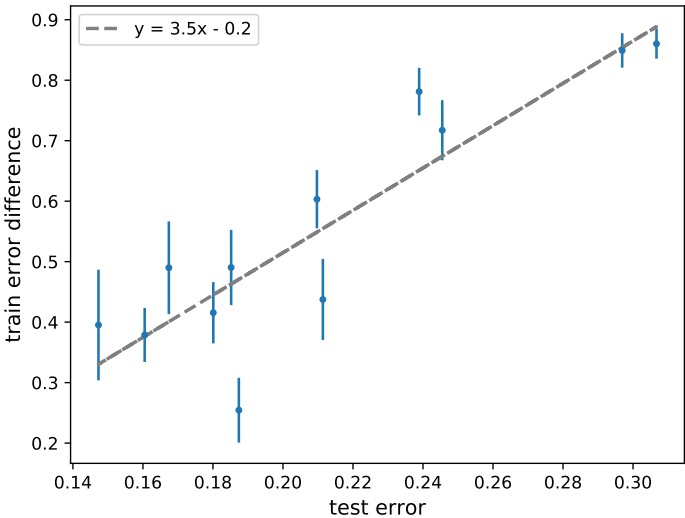

Figure 7: Train error difference $\delta E_{\text{train}}$ from Eq. (4) in function of test error for a fixed value of the perturbation intensity $\sigma = 0.5$, for minima obtained on the same VGG-like architecture and dataset (CIFAR-10) and with different values of dropout, batch size and weight decay. Each point shows the mean and standard deviation over 64 realizations of the perturbation. The models are taken from the public data of the PGDL competition.

## C    Deep networks experimental details

In this Section we describe in more detail the experiments reported in the Table 1 of the main text. In all experiments, the loss $\mathcal{L}$ is the usual cross-entropy and the parameter initialization is Kaiming normal. We normalize images in the train and test sets by the mean and variance over the train set. We also apply random crops (of width $w$ if image size is $w \times w$, with zero-padding of size 4 for CIFAR and 8 for Tiny ImageNet) and random horizontal flips. In

the following we refer to the latter procedure by the name "standard preprocessing". All experiments are implemented using PyTorch (Paszke et al., 2019).

For the experiments with eSGD and rSGD, we use the same settings and hyper-parameters used for SGD (unless otherwise stated and apart from the hyperparameters specific to these two algorithms).

For rSGD and unless otherwise stated, we set $y = 3$, $K = 10$ and use the automatic exponential focusing schedule for $\gamma$ reported in the main text.

For eSGD, we use again an exponential focusing protocol. In some experiments, we use a value of $\gamma_0$ automatically chosen by computing the distance between the configurations $w'$ and $w$ after a loop of the SGLD dynamics (i.e. in the first $L$ steps with $\gamma = 0$) and setting $\gamma_0 = \mathcal{L}(w)/d(w', w)$. Unfortunately, this criterion is not robust. Therefore, in some experiments the value of $\gamma_0$ was manually tuned. However, we found that eSGD is not sensitive to the precise value but to the order of magnitude.

We choose $\gamma_1$ such that $\gamma$ is increased by a factor of 10 by the end of the training. Unless otherwise stated, we set the the number of SGLD iterations to $L = 5$, SGLD noise to $\epsilon = 10^{-4}$ and $\alpha = 0.75$. Moreover, we use 0.9 Nesterov momentum and weight decay in both the internal and external loop. As for the learning rate schedule, when we rescale the total number of epochs for eSGD and rSGD, we use a rescaled schedule giving a comparable final learning rate and with consequently rescaled learning rate drop times as well.

## C.1 CIFAR-10 AND CIFAR-100

**SmallConvNet** The smallest architecture we use in our experiments is a LeNet-like network LeCun et al. (1998):

$$Conv(5 \times 5, 20) \ - MaxPool(2) - \ Conv(5 \times 5, 50) - MaxPool(2) - Dense(500) - Softmax$$

Each convolutional layer and the dense layer before the final output layer are followed by ReLU non-linearities.

We train the SmallConvNet model on CIFAR-10 for 300 epochs with the following settings: SGD optimizer with Nesterov momentum 0.9; learning rate 0.01 that decays by a factor of 10 at epochs 150 and 225; batch-size 128; weight decay 1e-4; standard preprocessing is applied; default parameter initialization (PyTorch 1.3). For rSGD we set $lr = 0.05$ and $\gamma_0 = 0.001$. For eSGD, we train for 60 epochs with: $\eta = 0.5$ that drops by a factor of 10 at epochs 30 and 45; $\eta' = 0.02$; $\gamma_0 = 0.5$; $\gamma_1 = 2 \cdot 10^{-5}$.

**ResNet-18** In order to have a fast baseline network, we adopt a simple training procedure for ResNet-18 on CIFAR-10, without further optimizations. We train the model for 160 epochs with: SGD optimizer with Nesterov momentum 0.9; initial learning rate 0.01 that decays by a factor of 10 at epoch 110; batch-size 128; weight decay 5e-4; standard preprocessing.

For rSGD we set $K = 1$ and learning rate 0.02. For eSGD, we train for 32 epochs with initial learning rate $\eta = 0.25$ that drops by a factor of 10 at epochs 16 and 25; $\eta' = 0.01$. In the case in which we drop the learning rate at certain epochs, we notice that it is important not to schedule it before that the training error has reached a plateau also for eSGD and rSGD.

**ResNet-110** We train the ResNet-110 model on CIFAR-10 for 164 epochs following the original settings of He et al. (2016): SGD optimizer with momentum 0.9; batch-size 128; weight decay 1e-4. We perform a learning rate warm-up starting with 0.01 and increasing it at 0.1 after 1 epoch; then it is dropped by a factor of 10 at epochs 82 and 124; standard preprocessing is applied.

For both eSGD and rSGD, we find that the learning rate warm-up is not necessary. For rSGD we set $\gamma_0 = 5e-4$. For eSGD, we train for 32 epochs with initial learning rate $\eta = 0.9$ that drops at epochs 17 and 25, SGLD learning rate $\eta' = 0.02$ and we set $\gamma_0 = 0.1$ and $\gamma_1 = 5 \cdot 10^{-4}$.

**PyramidNet+ShakeDrop** PyramidNet+ShakeDrop (Han et al., 2016; Yamada et al., 2018), together with AutoAugment or Fast-AutoAugment, is currently the state-of-the-art on CIFAR-10 and CIFAR-100 without extra training data. We train this model on CIFAR-10 and CIFAR-100 following the settings of Cubuk et al. (2018); Lim et al. (2019): PyramidNet272-$\alpha$200; SGD optimizer with Nesterov momentum 0.9; batch-size 64; weight decay 5e-5. At variance with Cubuk et al. (2018); Lim et al. (2019) we train for 300 epochs and not 1800. We perform a cosine annealing of the learning rate (with a single annealing cycle) with initial learning rate 0.05. ShakeDrop is applied with the same parameters as in the original paper (Yamada et al., 2018). For data augmentation we add to standard preprocessing AutoAugment with the policies found on CIFAR-10 (Cubuk et al., 2018) (for both CIFAR-10 and CIFAR-100) and CutOut (Devries & Taylor, 2017) with size 16.

For rSGD, we use a cosine focusing protocol for $\gamma$, defined at epoch $\tau$ by $\gamma_\tau = 0.5\gamma_{\max} \cos(\pi\tau/\tau_{\text{tot}})$, with $\gamma_{\max} = 0.1$. On CIFAR-10, we decrease the interaction step $K$ from 10 to 3 towards the end of the training (at epoch 220) in order to reduce noise and allow the replicas to collapse.

**EfficientNet-B0** EfficientNet-B0 is the base model for the EfficientNet family. In this section we train EfficientNet-B0 on CIFAR-100, starting from random initial conditions. We follow the same settings as Tan & Le (2019), with some differences: we train for 350 epochs with RMSprop optimizer with momentum 0.9; batch-size 64; weight decay 1e-5; initial learning rate 0.01 that decays by 0.97 every 2 epochs. We rescale image size to $224 \times 224$ and as data augmentation we apply standard preprocessing (with zero-padding of size 32) adding AutoAugment with the policies found on CIFAR-10 (Cubuk et al., 2018). For rSGD we set $\gamma_0 = 5e - 6$. For eSGD we used initial learning rate $\eta = 0.5$ that decays by 0.92 every 2 epochs and $\eta' = 0.05$.

## C.2 Tiny ImageNet

**ResNet-50** Entropic algorithms are effective also on more complex datasets. We train ResNet-50 on Tiny ImageNet (data downloaded from: "Tiny ImageNet Visual Recognition Challenge") for 270 epochs with: SGD optimizer with Nesterov momentum 0.9; initial learning rate 0.05 that decays by a factor of 10 at epochs 90, 180 and 240; batch-size 128; weight decay 1e-4. Standard preprocessing is applied together with Fast-AutoAugment with the policies found on ImageNet (Lim et al., 2019).

For eSGD we train the model for 50 epochs with $\eta = 0.8$ that drops by a factor of 10 at epochs 18, 36, 48 and $\eta' = 0.02$.

**DenseNet-121** For DenseNet-121 on Tiny ImageNet, the setting is the same as ResNet-50, except that we train the model for 200 epochs with learning rate drops at epochs 100 and 150.

For eSGD we train the model for 40 epochs with $\eta = 0.5$ that drops by a factor of 10 at epochs 25 and 30, $\eta' = 0.02$ and we set $\gamma_0 = 1.0$ and $\gamma_1 = 2 \cdot 10^{-5}$

