# OpenReview forum: "Entropic gradient descent algorithms and wide flat minima"
_ICLR.cc/2021/Conference — ICLR 2021 Poster_

### Official Review · AnonReviewer4 · 2020-10-28
**Ablation study on local entropy and local energy measures and how they correlate to generalization**

**Rating:** 5
**Confidence:** 4

**Review:**

Authors essentially study the generalization properties of networks trained via two different algorithms, Entropy SGD (eSGD) and Replicated SGD (rSGD)  against networks trained via SGD. Both eSGD and rSGD are algorithms that have been previously designed using the notion of entropy guided modified loss function. In this paper the authors compare the performance of broadly these three categories of networks in terms of Local Entropy (Eq. 3) and local energy (Eq. 4) (i.e. difference in training error when the weights are perturbed by factor \sigma), which was developed in [1]. With this the authors aim to find a correlation between networks with close to zero entropy and networks with lower local energy.

The first experiment uses a simple two layer neural network with fixed final layer weights and studies both local entropy and local energy as a function of weight perturbation. (I tried hard to find where Figure 1 is explicitly mentioned but I simply couldn't. I assume the experiments in Figure 1 correspond to the network eq (7) - please make this explicit).

Second set of experiments include simple classification test accuracies on deeper architectures such as ResNet110, DenseNet, EfficientNet and PyramidNet [this is not novel, just a replication and extension to newer architectures] (not including local energy, if I understand correctly.)

Third set of experiments are the main contribution of this paper - testing the correlation of local energy and local entropy - this is done on ResNet-18 trained on CIFAR10 dataset. While this does shed some light on the correlation of local entropy and local energy in promoting flatness, I'm not sure if one experiment on CIFAR10/Resnet18 is sufficient to justify correlations in deeper architectures that have been trained on larger datasets. What would have been more useful is to see if the same results translate for the models discussed in Section 6.1.

In conclusion, I do think this is an interesting correlation but not enough experiments are provided to justify this correlation.

References

[1] Fantastic Generalization Measures and Where to Find Them, Jiang et al ICLR 2019.

---

> ### Author Response · Authors · 2020-11-15
> **Response to AnonReviewer4**
>
> We are thankful for your comments. For a broad discussion of the novelty of our work, we refer to the general answer and respond to the other specific points you raised here:
>
> 1. We fixed the wrong reference to Figure 1 (erroneously called Figure 6) in the main text. As you point out, Figure 1 refers to Eq. 7. This is made explicit in the revised version of the manuscript, which we will soon upload.
> 2. On the insufficient number of experiments: In the Appendix, Sections B.2 and B.3, Figures 4, 5, 6 and 7 display various experiments on the other networks discussed in Section 6.1. This choice was dictated by space constraints. However, we could try to move some of those figures into the main text or perform further numerical experiments if the reviewer thinks that it would improve the paper.

---

### Official Review · AnonReviewer2 · 2020-10-30

**Rating:** 7
**Confidence:** 3

**Review:**

The paper studies (1) the relationship between the flatness of minima and their generalization properties, and (2) the connection between two measures of flatness, known as local entropy and local energy. Through a series of experiments, the authors show that the two measures are highly correlated and correlate well with generalization. They also empirically show that Entropy-SGD and Replicated-SGD, when used to explicitly optimize the local entropy, are able to flatter and better minima (in terms of lower generalization errors).

Clearly, the contributions of this work are not on either new algorithms or new measures, but rather it is a comprehensive empirical study that nicely brings seemingly different things together. I find the results interesting and the findings influential.

I have two clarifying questions:

(1) About activation functions in Section 5 and Section 6: For the results presented in Table 1, the deep networks use ReLU non-linearities, while in Section 5, the sign activation is shown in Equation 7 but then switched to an approximation by tanh(beta x). Can the authors clarify why using ReLU leads to an issue here? Please explain this “All these algorithms require a differentiable objective” and the use of a new loss function.

(2) Figure 3 and Figure 5, the training error difference of eSGD consistently decrease and gradually becomes smaller than that of SGD and rSGD. Does this have anything to do with the Langevin sampling? Can the authors give an explanation?

Minor comments:

+) In Equation (1), the loss L(w) is with respect to w, but what is L*(w, beta, gamma)? The same with d*(w, beta, gamma).

+) In Equation (3), E_train(w) is not defined. When \it{d}=0, then Omega(-d(w’, w)) in the denominator seems to be 0, which is strange?

+) Last paragraph before Section 6: do you mean Figure 1 instead of 6?. Also, there are some inconsistencies in the referencing figures: “Fig. 6”, “fig. 3” and “Figures 4 and 5”. The same problem is with equations: “(1)” vs “eq. (4)” and “Eq. (4)”.

---

> ### Author Response · Authors · 2020-11-15
> **Response to AnonReviewer2**
>
> We are thankful for your comments. For a broad discussion of the novelty of our work, we refer to the general answer and respond to the other specific points you raised here:
>
> 1. In the committee machine experiments, we use the sign activation function since it is the standard activation function used in the literature for the BP algorithm. Implementing the BP algorithm with different activation functions (such as ReLU) entails some additional complexity. The sign activation function has a zero derivative almost everywhere, which makes it hard to be used with gradient descent algorithms in general. To find solutions with sign activations with SGD we therefore replace them with tanh(beta*x) functions and increase beta during the training. This allows us to use a differentiable activation function that will, however, become a sign function in the limit beta to infinity. Notice that we could also have applied the commonly used straight-through estimator (STE) for binary activations with very similar results. The loss function that we define in Eq. 8 is the cross-entropy loss for binary classification, with an additional parameter omega that controls the growth of the weight norms.
>
> 2. The phenomenon seems to be connected to the specific training dynamics of eSGD, which is not clearly understood. While the two left panels from Figure 5 show that the flatness increases for all training algorithms, we note that in these panels the training has not yet finished, and it is not clear how comparable the curves for different training algorithms are. We agree that plotting these curves in the same plot can be misleading. We changed the plot accordingly and now compare in the different panels the same algorithms at different epochs or different algorithms at the end of training.  The updated manuscript will be upload shortly.
>
> Minor Comments:
>
> - L*(w, beta, gamma) and d*(w, beta, gamma) are the values that dominate the integral and which can be obtained by a saddle point approximation. We added this information to the manuscript, which we will upload shortly
> - E_train(w) is now defined.
> - Indeed, the expression in Eq. 3 is not strictly valid for d=0. However, the limit d->0 is well-defined: the numerator is always smaller or equal than the denominator and the argument of the log is always between 0 and 1. It tends to 1 for d -> 0 (since for almost any w, except for a set with null measure, there is always a sufficiently small neighborhood in which E_train is constant). We have now amended the explanation below the equation to clarify this issue.
> - Yes, we mean Figure 1 and not Figure 6 in last paragraph of Section 6. We apologize for the confusion. We fixed the referencing to figures and equations.
>
> EDIT: We made the 3rd bullet point of the response to the minor comments more precise.

---

### Official Review · AnonReviewer1 · 2020-10-31
**good paper but unclear what the contributions are**

**Rating:** 6
**Confidence:** 5

**Review:**

This paper studies local entropy measures for characterizing flat regions in the energy landscape of deep networks. The paper discusses, at length, two previously proposed algorithms named Entropy-SGD and Replicated-SGD and demonstrates, using (i) controlled experiments where Belief Propagation (BP) can be used to estimate the local entropy integral precisely, and (ii) empirical results on deep networks that flatter minima generalize better.

This paper presents a systematic analysis of the hypothesis that flat minima generalize better. This has been the subject of much debate in the recent literature because flatness is often characterized using spectral norm of the Hessian. The paper revisits the line of work that initiated this debate and shows that flatness, as measured by local entropy instead, indeed correlates with good generalization.

The paper lacks in terms of novelty in the sense that it does not propose a “new” algorithm, but the reviewer believes that not all publications need to. A paper like this with carefully constructed constructed and a clear conclusion is also a valuable addition to this debate. I am however concerned about the incremental value of this manuscript in view of the long line of work that it builds upon. I am recommending a weak accept but I am willing to increase my score if the authors make a convincing case against this concern.

Comments.

1. The paper, as noted above, heavily builds upon existing work, in particular series of works of Baldassi et al. and Chaudhari et al. The papers introduce local entropy, focussing, replicated SGD, BP for calculating the local entropy etc. Entropy-SGD/Parle were also shown to work well for state-of-the-art deep networks then. The authors should state clearly what the concrete contributions of this manuscript are.
2. I like the section on measuring flatness for shallow architectures using BP. It would be good to include the BP calculations in the Appendix.
3. The idea of making the cross-entropy loss invariant to the scale of the weights by keeping them normalized can also be used with deep networks using weight normalization https://arxiv.org/abs/1602.07868.
4. Can you characterize the shape of the wide region? I believe this will lend more insight in why wide minima generalize better for deep networks.

---

> ### Author Response · Authors · 2020-11-15
> **Response to AnonReviewer1**
>
> We are thankful for your comments. For a broad discussion of the novelty of our work, we refer to the general answer and respond to the other specific points you raised here:
>
> 1. Please see the general answer.
> 2. The BP calculations on the one-hidden layer architecture are rather lengthy and have been recently detailed in Appendix IV of Ref. [1]. We updated the manuscript (which will uploaded shortly) outlining the computation scheme, adding some further details specific to our work, and pointing to that reference. Extending this technique to DNNs is an ongoing line of our research.
> 3. Indeed, it is certainly possible and it would be an interesting avenue to explore next. For the purposes of the present manuscript, the need to control the norms arises mainly from the requirement of drawing a meaningful comparison between different solutions. This is crucial when measuring the local entropy, which is not norm-invariant. It is also a source of confusion in discussions about the relation between flatness and generalization, see for example Dinh et al. [2]. For the tests on DNNs, however, we have only directly measured the flatness using the local energy, which uses a multiplicative noise and therefore does not suffer from this issue. Furthermore, the baseline models that we used did not apply weight normalization, and therefore we followed the previous SOTA for those architectures in this regard. We have now added a brief comment about this in the manuscript, along with the suggested citation.
> 4. Thanks for the suggestion. This is a very important question which is, however, not easy to answer. We are exploring different methods to characterize the shape of the landscape around the minima found by different algorithms, but there are several subtleties, and we consider this to be beyond the scope of this paper. We note that analytical calculations indicate that the shape of the landscape is very complex even in shallow systems and it is not clear whether they even have a characteristic scale [3].
>
> [1] Baldassi et al. , “Shaping the learning landscape in neural networks around wide flat minima”, PNAS 2019.
>
> [2] Dinh, Laurent, et al. "Sharp minima can generalize for deep nets." arXiv preprint arXiv:1703.04933 (2017).
>
> [3] Baldassi, Carlo, et al. "Subdominant dense clusters allow for simple learning and high computational performance in neural networks with discrete synapses." Physical review letters 115.12 (2015): 128101.

---

### Official Review · AnonReviewer3 · 2020-11-05
**Official Blind Review #3**

**Rating:** 6
**Confidence:** 4

**Review:**

Update after response: I appreciate the authors making their contributions clearer, and adding details about the training loss and error. I have increased my score accordingly.

Original Review:

This paper presents an empirical evaluation of whether flatness correlates with generalization using a few different definitions of flatness - local energy and local entropy. The authors study two training procedures - entropy sgd and replicated sgd, and show that deep networks trained using these procedures are able to locate flatter solutions that also generalize better.

While the paper presents some interesting empirical confirmation of the correlation between local entropy/energy and generalization, the algorithms presented in this paper have also been defined in previous work, and this phenomenon has been repeatedly observed with different definitions of flatness [1,2]. The authors also do not present any reasons to expect that generalization is related to flatness that are grounded in theory. The contributions of this paper thus seem to be confirmation of previously observed phenomena [3,4].

Moreover, in the experiments it is not clear whether the solutions that are reached by the training procedures actually correspond to local or global minima of the cross-entropy loss function (There is also the issue that minimizers of the cross-entropy loss function occur at infinity). The authors do not report the training loss at the solutions that they choose to plot, and the training error also does not seem to be zero (plots corresponding to Figure 3). I am not sure whether it makes sense to call these solutions minima, and compare their flatness.


[1] Neyshabur, B., Bhojanapalli, S., McAllester, D., & Srebro, N. (2017). Exploring generalization in deep learning. In Advances in neural information processing systems (pp. 5947-5956).

[2] Keskar, N. S., Mudigere, D., Nocedal, J., Smelyanskiy, M., & Tang, P. T. P. (2016). On large-batch training for deep learning: Generalization gap and sharp minima. arXiv preprint arXiv:1609.04836.

[3] Baldassi, C., Pittorino, F., & Zecchina, R. (2020). Shaping the learning landscape in neural networks around wide flat minima. Proceedings of the National Academy of Sciences, 117(1), 161-170.

[4] Chaudhari, P., Choromanska, A., Soatto, S., LeCun, Y., Baldassi, C., Borgs, C., ... & Zecchina, R. (2019). Entropy-sgd: Biasing gradient descent into wide valleys. Journal of Statistical Mechanics: Theory and Experiment, 2019(12), 124018.

---

> ### Author Response · Authors · 2020-11-15
> **Response to AnonReviewer3**
>
> We are thankful for your comments. For a broad discussion of the novelty of our work, we refer to the general answer and respond to the other specific points you raised here:
>
> > The authors also do not present any reasons to expect that generalization is related to flatness that are grounded in theory.
>
> Entropic algorithms are based on the local entropy theoretical framework, which analytically showed on simplified models that wider minima generalize better than shallow ones [1]. One of the main points of our manuscript is to numerically validate these theoretical predictions on current deep architectures.
>
> > The contributions of this paper thus seem to be confirmation of previously observed phenomena [3,4].
>
> Please see general answer.
>
> >Moreover, in the experiments it is not clear whether the solutions that are reached by the training procedures actually correspond to local or global minima of the cross-entropy loss function . (There is also the issue that minimizers of the cross-entropy loss function occur at infinity).
>
> Thanks to the focusing procedure, at the end of the training our objective functions reduce to the original cross-entropy, therefore the configurations we find are (approximate) minimizers of the original loss.  In our work, we optimized the networks in our experiments until an almost stationary and low value of train error/loss was found, using standard learning-rate-drop schedules. We make this clearer in the revised version of the manuscript. If the comment concerns the general question of local versus global minima: In general, it is difficult to know whether the minima found in neural networks are local or global.  There is some amount of evidence that in overparametrized networks algorithmically accessible (if not all) minima are global minima (within some tolerance). Moreover, the Hessian computed at the endpoint of SGD-based training typically exhibits a large number of flat directions and a few slightly negative ones [2]. Therefore, one should refer to the learning outcomes as quasi-minima.
>
> >The authors do not report the training loss at the solutions that they choose to plot,
>
> We will report the train loss as suggested in the updated manuscript, which will be uploaded shortly.
>
> > and the training error also does not seem to be zero (plots corresponding to Figure 3). I am not sure whether it makes sense to call these solutions minima, and compare their flatness.
>
> Similar to [3], we consider configurations with very low training error (in [1], configurations with training error < 1% are compared). While this does not guarantee that we are exactly at a minimum, also due to the considerations above, we assume that this condition brings us at least to a point where it is sensible to compare flatness. Let us also stress that in the ResNet-110 case, where the training error is very close to zero for all algorithms, the picture holds as well.
>
> [1]  Baldassi et al. “Unreasonable effectiveness of learning neural networks”, PNAS (2016)
>
> [2] Sagun et al. “Empirical Analysis of the Hessian of Over-Parametrized Neural Networks”, ICLR 2018
>
> [3] Jiang, Yiding, et al. "Fantastic generalization measures and where to find them." arXiv preprint arXiv:1912.02178 (2019).

---

### Author Response · Authors · 2020-11-15
**General Answer for all Reviewers**

We thank all reviewers for the detailed and constructive criticisms and comments on our work. We will provide detailed answers to their specific points below. Here, we will respond to the common concern raised about the novelty of the results. We agree that this point could have been made clearer in the first version of the manuscript.  We are thus providing clarifications here and in the revised version of the manuscript, which we will upload shortly.

While eSGD and rSGD have been introduced in previous works, [1] and [2] respectively (as quoted in the manuscript), we believe that our manuscript represents a valuable addition to the current knowledge considering the following points:
1. rSGD has never been implemented in the deep neural networks setting, and its generalization properties have not been studied before. In [2], only shallow neural networks (committee machines) with binary weights trained on random (non-generalizable) patterns have been tested. Thus, our manuscript presents the first detailed report on the properties of rSGD in a realistic setting. EASGD (Zhang et al. (2014)), which has some common features with rSGD, was designed to deal with distributed scenarios and not for finding wide minima to improve generalization. In fact, the absence in EASGD of the focusing procedure makes it not very effective for the geometrical objective of finding wide minima.
2. eSGD has been introduced in [1] and applied to image classification tasks. However, the paper does not report a consistent improvement in the generalization performance compared to standard SGD training for image classification.
3. To the best of our knowledge, our work is the first one reporting improved generalization performances in DNN for eSGD and rSGD in image classification. Besides the potential relevance for applications, we also believe that this is important for the theoretical understanding of DNN to have some clear numerical results (separate from all the usual heuristics used to improve generalization) which point to a well-defined geometrical interpretation.
4. We obtained these improvements by designing novel simulation protocols for both algorithms (with minimal hyperparameter tuning involved). We have also made the code for both algorithms freely available (supplementary material) and thereby hope to foster research and applications of entropic algorithms. Clearly, a systematic optimization of the hyperparameters would result in further improvements.
5. Our work shows a correlation between two a-priori distinct measures of flatness, namely local entropy and local energy, and show that both correlate with generalization performance in the settings we tested. For shallow networks, we also have a semi-analytic method based on Belief Propagation that can be used to estimate the entropy within an extensive region around the minimizers. With some (considerable) effort, this tool could be extended to DNN.
6. We measure both generalization and flatness on a variety of state-of-the-art architectures not investigated by the “entropic” literature and show that the picture holds for modern deep neural networks as well. The correlation between flatness and generalization is a topic of ongoing debate and no consensus has been reached. We believe that our work adds evidence to this debate, by analyzing algorithms that optimize a clear objective related to the geometry of the loss landscape.

To summarize, while we do not present completely new algorithms, we do present a series of modifications that are instrumental in making those algorithms generalize well with minimum hyperparameter tuning. Moreover, we show that entropic algorithms improve generalization also in the case of modern architectures and realistic datasets. We use this context to shed light on the question of the correlation between generalization and flatness, showing that entropic algorithms indeed lead to increased flatness in this regime and furthermore show a connection between different measures of flatness.

[1] Chaudhari et al.  “Entropy-sgd: Biasing gradient descent into wide valleys”  ICLR (2017) and Journal of Statistical Mechanics: Theory and Experiment, 2019

[2]  Baldassi et al. “Unreasonable effectiveness of learning neural networks”, PNAS (2016)

---

### Decision · Program_Chairs · 2021-01-07
**Final Decision**

**Decision:**

Accept (Poster)

**Comment:**

This paper studies the link between generalization behavior and "flatness" of the loss landscape in deep networks. Specifically, the authors study two measures of flatness (local entropy and local energy), and show that these two measurements are strongly correlated with one another. Moreover they show via a careful set of numerical experiments that two previously proposed algorithms (entropy SGD and replica SGD) that optimize for local entropy tend to both find flatter minima as well as provide better generalization.

Despite the fact that the paper proposes no new models or algorithms, the experiments are compelling and provide non-trivial insights into predicting generalization behavior of deep networks, as well as solid evidence on the benefits of entropy regularization in SGD. The authors also seem to have satisfactorily answered the (numerous) initial concerns raised by the authors. Overall, I recommend an accept.